# Interfacial Microstructure and Mechanical Reliability of Sn-58Bi/ENEPIG Solder Joints

Cheng Chen [1,*], Cheng Wang [1], Huhao Sun [1], Hongbo Yin [1], Xiuli Gao [1], Hengxu Xue [1], Dahai Ni [1], Kan Bian [2] and Qilin Gu [3,*]

1 China State Shipbuilding Corporation, Limited (CSSC) 723rd Research Institute, Yangzhou 225001, China; nuaawc@163.com (C.W.); sunhuhao2012@163.com (H.S.); yinhongbo2004@126.com (H.Y.); yzgxl@163.com (X.G.); xhx08648@126.com (H.X.); seuseayou@outlook.com (D.N.)
2 Nanjing Institute of Technology, School of Materials Science and Engineering, Nanjing 211100, China; bk@nuaa.edu.cn
3 National Engineering Research Center for Special Separation Membrane, Nanjing Tech University, Nanjing 210009, China
* Correspondence: chencheng5670@163.com (C.C.); gu_qilin@nuaa.edu.cn (Q.G.)

**Abstract:** The 42 wt.% Sn–58 wt.% Bi (Sn-58Bi) Ball Grid Array (BGA) solder balls were mounted to electroless nickel-electroless palladium-immersion gold (ENEPIG) pads by employing the reflow process profile. The effects of reflow cycles and aging time on the interfacial microstructure and growth behavior of intermetallic compounds, as well as the mechanical properties, were investigated. Pd-Au-Sn intermetallic compound (IMC) was formed at the Sn-58Bi/ENEPIG interface. With the increase in reflow cycles and aging time, the IMC grew gradually. After five reflow cycles, the shear strength of the Sn-58Bi/ENEPIG solder joints first decreased and then increased. After 500 h of aging duration under −40 °C, the shear strength of the Sn-58Bi/ENEPIG solder joints decreased by about 12.3%. The fracture mode transferred from ductile fracture to ductile and brittle mixed fracture owing to the fact that the fracture location transferred from the solder matrix to the IMC interface with the increase in reflow cycles and aging time.

**Keywords:** electroless nickel-electroless palladium-immersion gold; Sn-58Bi; ball grid array; intermetallic compound

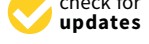



## 1. Introduction

The eutectic Sn-37Pb solder is an important material in the field of electronic packing because of its excellent wettability, electronic conductivity and mechanical properties [1,2]. However, due to the threat of lead to the human body and the environment, lead-free solder has become the development direction in the electronic industry with the restriction of hazardous substances (RoHs) [3,4]. Sn–Ag, Sn–Zn, Sn–Ag–Cu and Sn–Bi alloys are developed in order to reach or approach the properties of the eutectic Sn-37Pb solder [5–8]. Eutectic Sn-58Bi solder with a melting point of 138 °C has been considered a promising lead-free solder for low-temperature applications, especially in the fields of MCM (Multichip Module) and 3D heterogeneous integration modules. This solder possesses good creep resistance and high tensile strength [9]. In addition, it can avoid warpage in printed circuit boards (PCBs) and electronic components due to thermal mismatch during the soldering process [10].

On the other hand, during PCB manufacturing, a solderable finish is plated over the Cu substrate. At present, electroless nickel-immersion gold (ENIG) has been widely used as a solderable finish in the electronic industry due to its good wettability [11–13]. During the soldering process, the intermetallic compound (IMC) can be formed at the solder/PCB finish interface, which is essential for mechanical, conductive and thermal properties. As for the ENIG finish, the electroless nickel layer acts as a diffusion barrier against Cu migration,

and the immersion gold layer can prevent surface oxidation to improve weldability [14]. However, during the immersion gold plating process, ENIG corrodes the Ni (P) layer and causes cracks, which are called "black pad" phenomena [15,16]. Recently, a new solderable finish called ENEPIG finish has emerged [17,18]. A thin and dense electroless palladium layer is added between the Ni (P) layer and Au layer, which can prevent the oxidation reaction of Ni (P) and the "black pad" phenomenon thoroughly [19]. Additionally, ENEPIG finish shows better weldability, corrosion resistance and reliability [20,21].

Studies on the behavior of IMC chemical reactions with ENIG and ENEPIG have been well documented. Yoon et al. [18] found that needle-shaped (Cu, Ni) $_6Sn_5$ IMC and P-rich Ni layer formation deteriorated the shear strengths of the ENEPIG joints. Tian et al. [22] found that the shear strength of the SAC305/ENEPIG joints was consistently higher than that of the SAC305/ENIG joints due to the slightly slow growth rate of interfacial IMCs. With the increase in thermal shock cycles, the fracture mode of the SAC305/ENIG joints switched from ductile to ductile–brittle mixed fracture mode, while the fracture mode of the SAC305/ENEPIG joints was consistently ductile. Chi et al. [23] found two layers of IMCs with compositions of $(Au_{0.30}Ni_{0.70})(Sn_{0.90}Bi_{0.10})_4$ and $Ni_3Sn_4$ were formed at the Sn-58Bi solder/ENIG finish interface of the aged joints, and the relationship between ball shear strength($S$) and thickness($X$) of $(Au_{0.30}Ni_{0.70})(Sn_{0.90}Bi_{0.10})_4$ IMC layer: $S = 7.13 - 0.33X$ was researched. Chen et al. [24] only detected NiSn IMC at the Sn-58Bi/Ni interface and found that the growth of IMC followed diffusion-controlled kinetics. Kim et al. [25] compared the effects of phosphorous and pure Pd in a thin ENEPIG surface finish on the interfacial reactions and mechanical properties of the Sn-58Bi/thin-ENEPIG solder joints and found that the Sn-58Bi solder with phosphorous in the finish has a higher reliability than that with pure Pd in the finish.

The Sn-58Bi solder joint is applicable for the field of MCM (Multichip Module) and SIP (System in Package) modules. According to the National Military Standard of the People's Republic of China (GJB 8481-2015), General Specification for Microwave Assembly, the working temperature of these modules is in the range of −40 °C to 70 °C. These modules may undergo multiple reflow cycles during rework or multi-gradient assembly [26] and will be in service at temperatures as low as −40 °C for a long time. Therefore, a detailed study on the interfacial reactions and growth behavior of IMC, as well as the shear strength and fracture mode of the Sn-58Bi/ENEPIG solder joints with different reflow cycles and aging at a temperature as low as −40 °C are presented in this work.

## 2. Materials and Methods

### 2.1. Experimental Materials

A PCB made of flame-retardant-4 (FR4) with an ENEPIG plating process was used as the substrate. The ENEPIG process included electroless nickel plating, electroless palladium plating, and immersion gold plating, successively. The electroless nickel plating process was a redox reaction under the action of metal catalysis. Hypophosphite was used as a reducing agent in the electroless nickel plating process, and the reaction process includes:

$$[H_2PO_2]^- + H_2O \rightarrow [HPO_3]^{2-} + H^+ + 2H, \tag{1}$$

$$Ni^{2+} + 2H \rightarrow Ni\downarrow + 2H^+, \tag{2}$$

$$2[H_2PO_2]^- + H \rightarrow [HPO_3]^{2-} + H_2O + P + H_2\uparrow. \tag{3}$$

Electroless palladium plating process was also an autocatalytic redox reaction procedure with hypophosphite as a reducing agent. The reaction process includes:

$$[H_2PO_2]^- + H_2O \rightarrow [HPO_3]^{2-} + H^+ + 2H, \tag{4}$$

$$Pd^{2+} + 2H \rightarrow Pd\downarrow + 2H^+, \tag{5}$$

$$2[H_2PO_2]^- + H \rightarrow [HPO_3]^{2-} + H_2O + P + H_2\uparrow. \tag{6}$$

The immersion gold plating process was a displacement process where Au and Pd were replaced to obtain the Au layer. The reaction process was:

$$2Au^+ + Pd \rightarrow 2Au + Pd^{2+}. \tag{7}$$

The thickness of Ni (P), Pd and Au layers was about 5 μm, 0.1 μm and 0.05 μm, respectively. The PCB pad opening size was 0.4 mm. After being dipped in Sn-58Bi solder paste (HL-6RC-4258, Honglian, Suzhou, China), Sn-58Bi BGA solder balls (QW-SB42-Q500, Qwin, Chongqin, China) of 0.5 mm in diameter were mounted to ENEPIG pads by employing the reflow process profile. After reflowing and cooling to room temperature, the soldering samples were cleaned to remove the flux residuals by manual spraying cleaning agents (VIGON EFM, ZESTRON, Ingolstadt, Germany). Figure 1 shows a schematic diagram of the Sn-58Bi/ENEPIG solder joints. The reflow process was realized by a reflow oven (VXC 421, Rehm, Blauboylen, Germany) with six heating zones and two cooling zones. The temperature of the heating zones was measured by a thermodetector, which is shown in Table 1. The thermodetector displayed the temperature profile for the reflow process of Sn-58Bi/ENEPIG solder joints, and the result is shown in Figure 2. The peak temperature ($T_{max}$), the melting point ($T_e$), the preheating time and the cooling rate was about 170 °C, 138 °C, 180–220 s, 60–90 s and 2–5 °C/s, respectively.

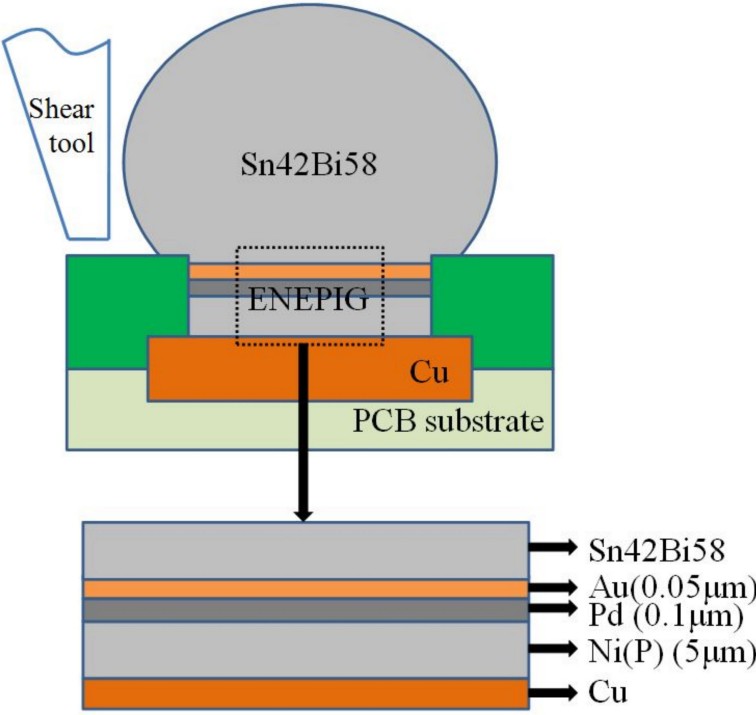

**Figure 1.** Schematic diagram of the Sn-58Bi/ENEPIG solder joints.

**Table 1.** Temperature of the heating zones.

| Zone<br>Temperature | 1 | 2 | 3 | 4 | 5 | 6 |
|---|---|---|---|---|---|---|
| Upper furnace/°C | 120 | 130 | 135 | 140 | 160 | 195 |
| Bottom furnace/°C | 120 | 130 | 135 | 140 | 160 | 195 |

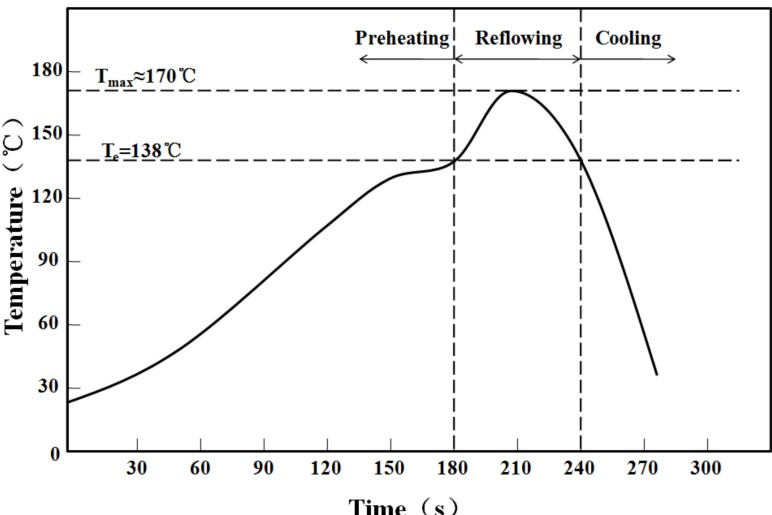

**Figure 2.** Temperature profile for the reflow process of Sn-58Bi/ENEPIG solder joints.

### 2.2. Aging Test

The aging test at −40 °C ranging from 100 h to 500 h was performed on the reflowed joints by using a rapid temperature change test chamber (SU-662, Espec, Osaka, Japan). In addition, the Sn-58Bi/ENEPIG solder joints were conducted using a reflow oven under different reflow cycles.

### 2.3. Microstructure Observation

After reflowing different cycles, as well as aging under different process parameters, the samples were mounted, inserted, ground and polished successively to observe the cross-sectional microstructure. The interfacial microstructure of the aged solder joints was observed by a scanning electron microscope (SEM, S-4800, Hitachi, Tokyo, Japan) and the composition of IMC, solder and finish was analyzed by energy-dispersive spectroscopy (EDS). The interfacial IMC thickness was calculated using the following equation:

$$L_{IMC} = (N_{IMC}/N_{SEM}) \times L_{SEM} \tag{8}$$

where $L_{IMC}$ was the interfacial IMC thickness, $L_{SEM}$ was the height of the SEM image, and $N_{IMC}$ and $N_{SEM}$ were the number of pixels for the interfacial IMC and the entire SEM image, respectively. The values of $N_{IMC}$ and $N_{SEM}$ were obtained using Image J software. For each test condition, at least five SEM images taken at different locations were utilized to calculate the average value of the interfacial IMC thickness.

### 2.4. Ball Shear Test and Fracture Morphology Analysis

The ball shear strength test was performed using a global bond tester (DAGE4000, Nordson, Aylesbury, UK). The shear speed was 500 μm/s, and the shear height was set at about 50 μm. After the ball shear test, fracture morphology and composition were analyzed by SEM and EDS.

## 3. Results and Discussion

### 3.1. Interfacial Reactions

Figure 3 shows the cross-sectional image of Sn-58Bi/ENEPIG solder joints after reflowing and the enlarged view of the white square area is presented in Figure 4. In combination with the Sn-58Bi phase diagram, the Sn-58Bi solder ball contained an eutectic lamellar microstructure of β-Sn solid solution (white color) and Bi-rich phase (gray color) [22]. A continuous flake-type IMC layer with a thickness of 1–3 μm was formed at the Sn-58Bi/ENEPIG interface. In addition, it can be seen that a P-rich Ni layer was beneath

the interfacial IMC layer. EDS was further carried out to analyze the interfacial reactions between the Sn-58Bi solder ball and ENEPIG finish. According to Figures 5 and 6 results, Pd and Au elements in the ENEPIG finish with the Sn element in Sn-58Bi solder were formed at the Sn-58Bi/ENEPIG interface. Additionally, it can be seen that during the reflow process, most of the Pd element was dissolved into the molten solder to react at the interface. The remaining thin Pd layer prevented the diffusion of the Ni layer, which could be evidenced by the results in Figure 5. The chemical composition of the interfacial IMC was 1.82 at. % Ni, 14.78 at. % Pd, 74.36 at. % Sn, and 9.04 at. % Au. As can be seen, the atomic proportion of Pd and Au is not 1:1 in the IMC, which means Au has replaced some of the Pd atoms in $PdSn_4$. At the same time, the atomic proportion of (Pd, Au) and Sn is 1:3.12. Considering the acceptable measurement error and the possible compound that three elements can form, we infer that it is most likely $(Pd, Au)Sn_4$. The results were coincident with those of a previous study [25].

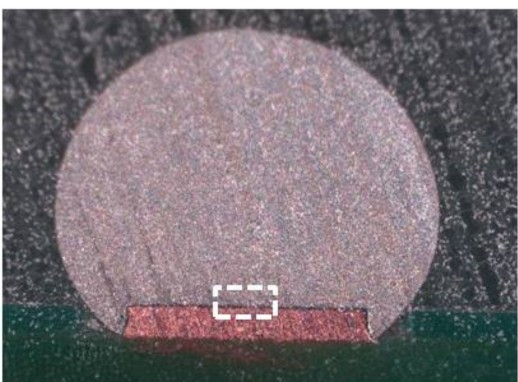

**Figure 3.** Cross-sectional images of Sn-58Bi/ENEPIG solder joints after reflowing.

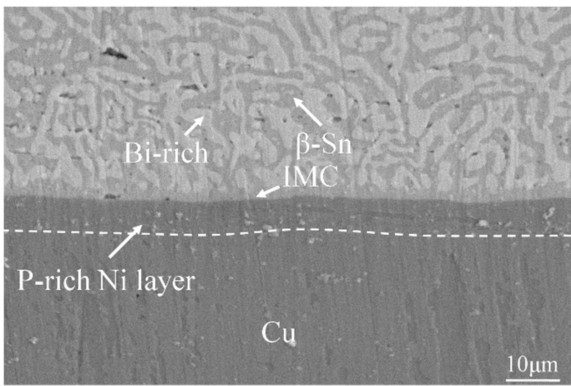

**Figure 4.** Cross-sectional SEM images of the white square in Figure 3.

*3.2. Growth Behavior of Interfacial IMC and Mechanical Properties for Different Reflow Cycles*

The reflow process of Sn-58Bi/ENEPIG solder joints was carried out for one to five cycles, and the surface morphology of solder joints is shown in Figure 7. It can be seen that after one reflow cycle, the solder joints were bright and had good wettability. After two reflow cycles, the surface of the solder joints was wrinkled. With the increase in reflow cycles, the wrinkle became obvious, the solder joint surface became black and the oxidation phenomenon was more serious. SEM images of Sn-58Bi/ENEPIG joints with reflow cycles ranging from one to five are shown in Figure 8. It was found that the thickness of the IMC layer gradually increased with the increase in reflow cycles. The thickness of the IMC layer formed at the interface for reflow cycles ranging from one to five was 1.7, 2.3, 3.2, 3.8 and 4.3 μm, respectively, as shown in Figure 9. By fitting the results, the relationship between reflow cycles (X) and thickness (Y, μm) of the IMC layer can be described as:

$Y = 0.67X + 1.05$. In addition, with the increase in reflow cycles, the Bi rich phase and β-Sn phase showed a coarse growth trend.

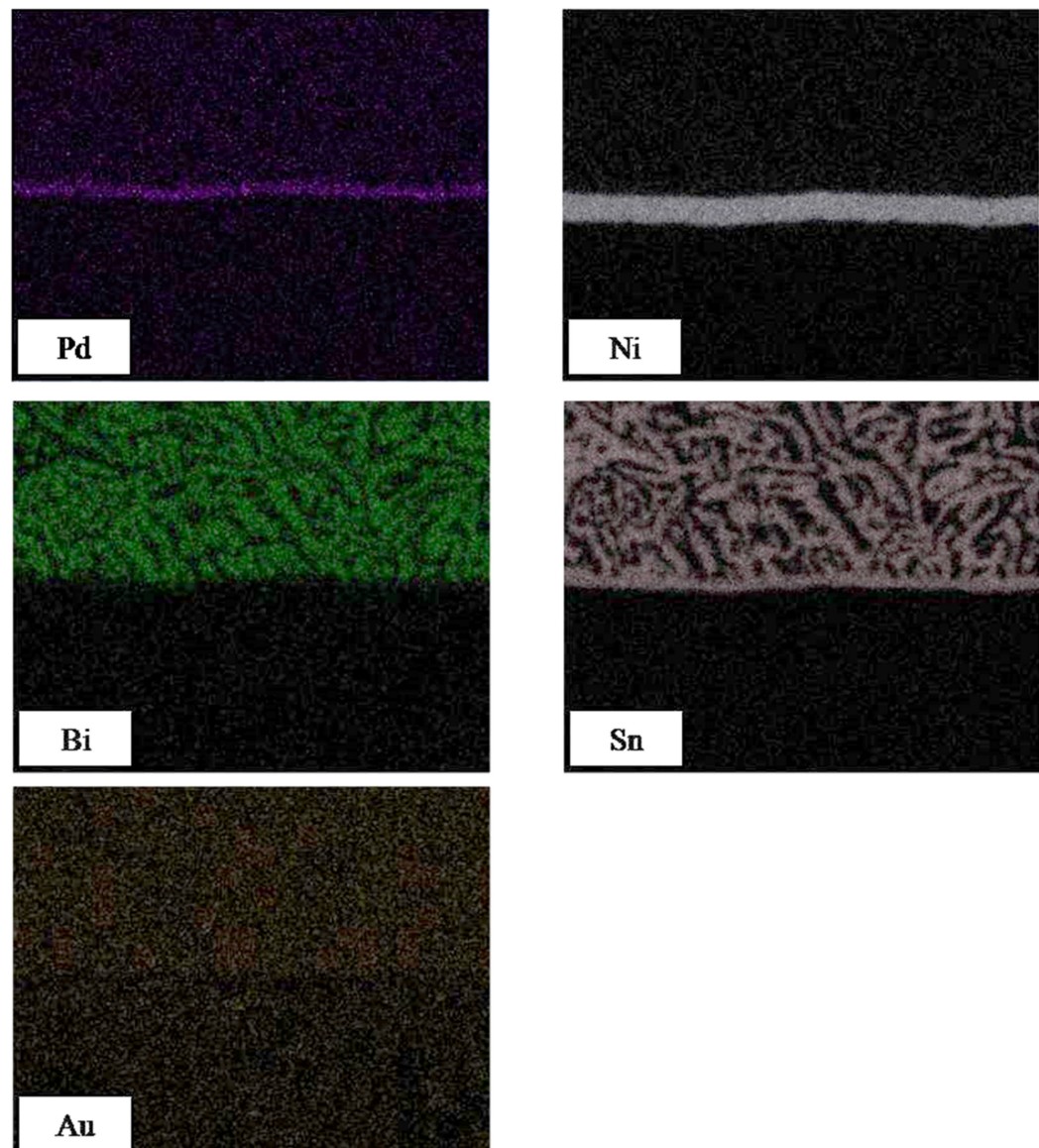

**Figure 5.** EDS results of Sn-58Bi/ENEPIG solder joints after reflowing.

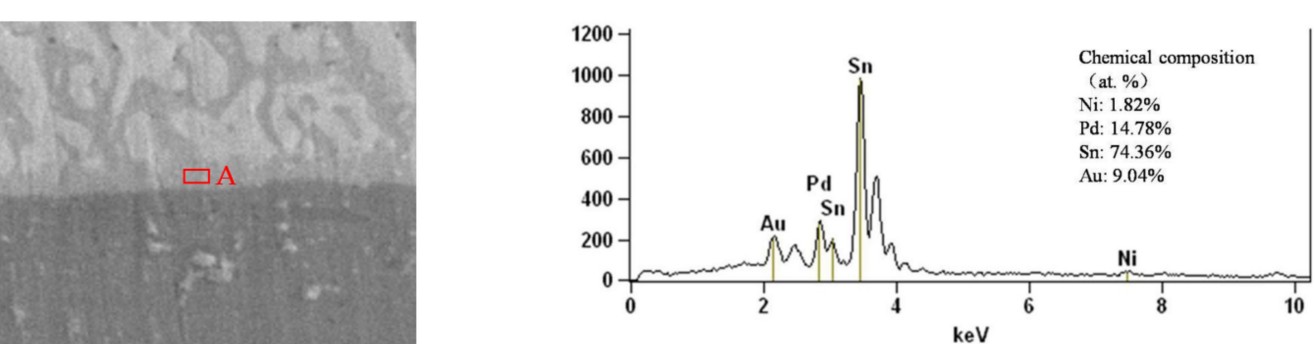

**Figure 6.** EDS result of the interfacial IMC in area A of enlarged Figure 4.

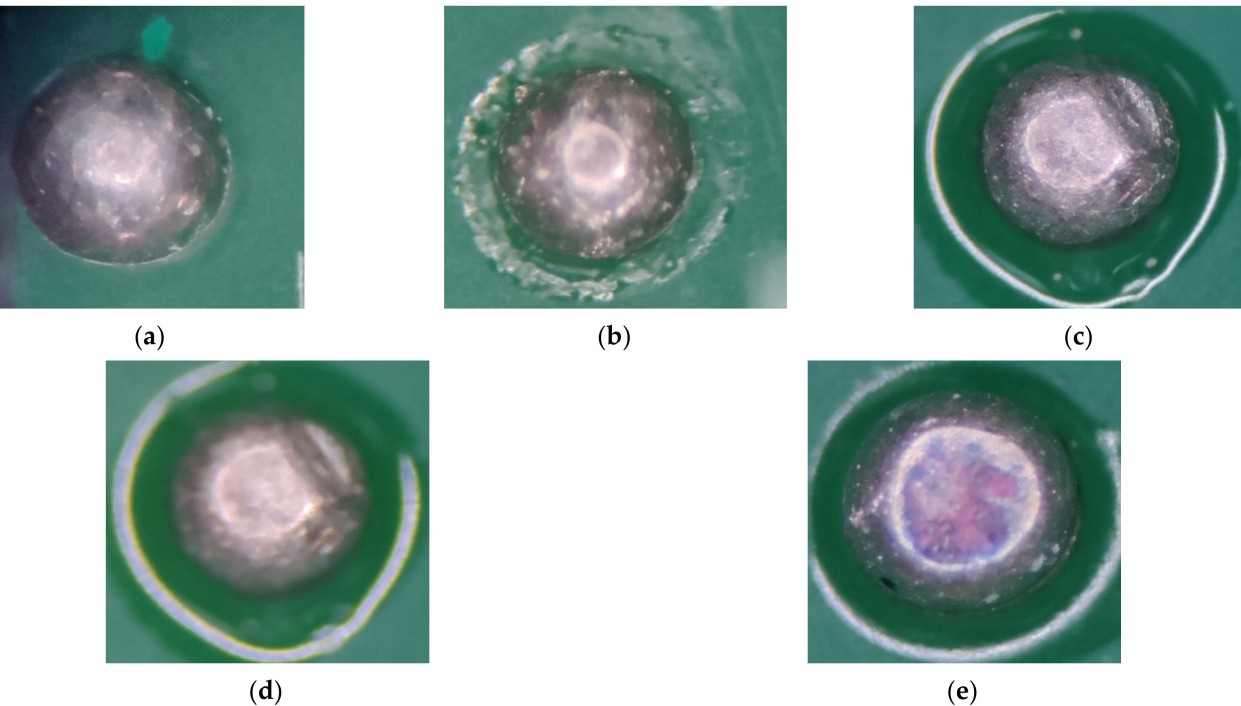

**Figure 7.** Surface morphology of solder joints with different reflow cycles: (**a**) one reflow cycle, (**b**) two reflow cycles, (**c**) three reflow cycles, (**d**) four reflow cycles and (**e**) five reflow cycles.

The ball shear test on solder joints with different reflow cycles was carried out. The ball shear strength of the solder joints for reflow cycles ranging from one to five was 86.15, 81.85, 66.00, 76.20 and 84.79 MPa, respectively. With the increase in reflow cycles, the ball shear strength first decreased and then increased. The shear strength of solder joints reached the lowest value of 66.00 MPa after three reflow cycles. However, the increase in reflow cycles had little effect on the shear strength of the solder joints. In addition, the shear strength of the solder joints was slightly higher than that reported in reference [25]. In the literature [25], the shear strength of the solder joints was about 80 MPa after one reflow cycle, while the shear strength of the solder joints in this work was 86.15 MPa. This was due to the different shear speeds of the solder joints in the measurement. Specifically, in the literature [25], the shear speed of the solder joints was 200 μm/s, while in this work, the shear speed of the solder joints was 500 μm/s. According to the National Military Standard of the People's Republic of China (GJB 7677-2012), Test Methods for Ball Grid Array (BGA), the shear strength of the solder joints increases with the increase of shear test speed. This can well explain the slightly higher shear strength of the solder joints in this work. The ball shear tested samples were analyzed by SEM and the result is shown in Figure 10. It was found that after two reflow cycles, the fracture surface consisted of a solder matrix. After three reflow cycles, the fracture surface consisted completely of an IMC layer, which also explained its lowest shear strength. After five reflow cycles, the fracture surface consisted of solder matrix and IMC layer, and the shear strength was improved compared with that of three reflow cycles. According to the analysis of SEM images, the fracture mode of one and two reflow cycles was ductile fracture, while the fracture of three reflow cycles was brittle fracture, and the fracture of four and five reflow cycles was mixed with ductile and brittle fracture. Therefore, in practical application, the number of reflow cycles shall not exceed three.

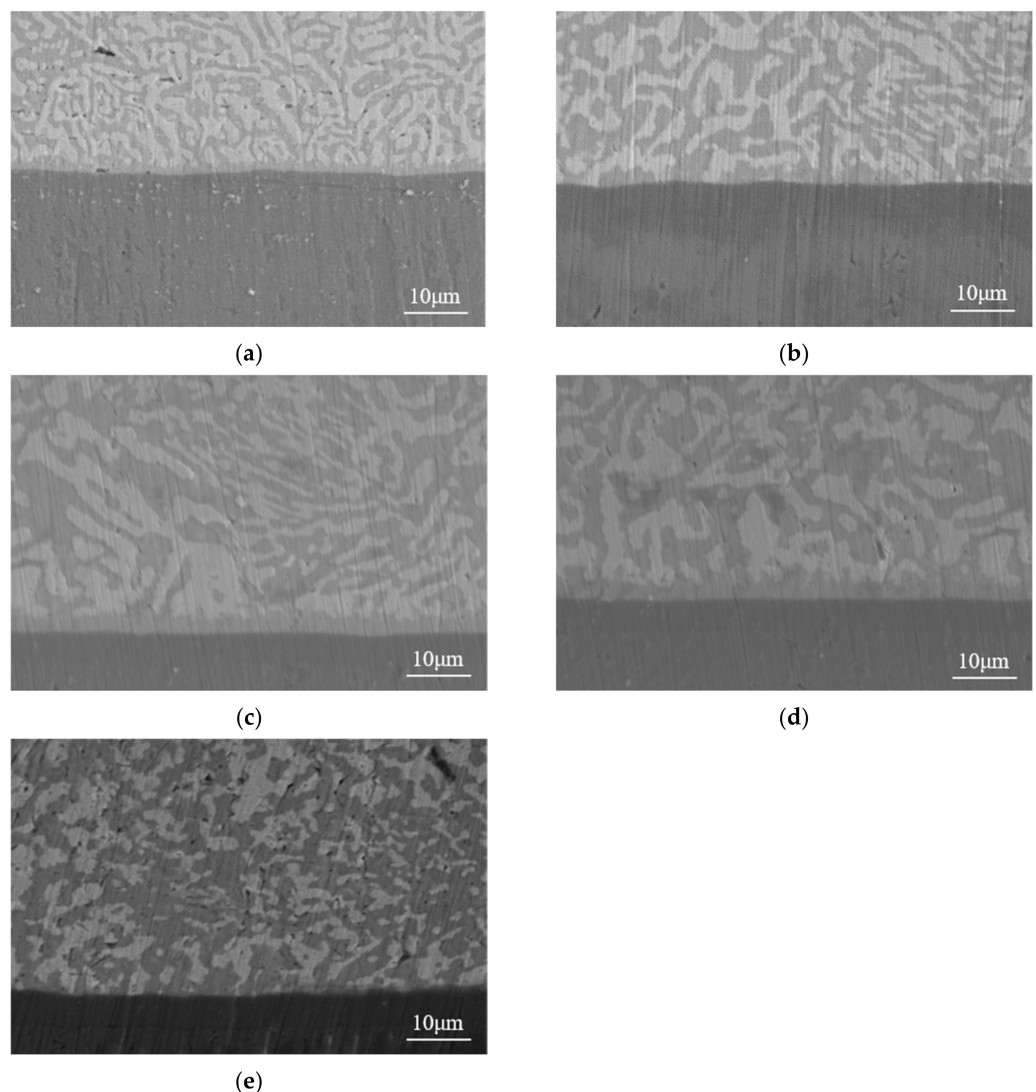

**Figure 8.** Cross-sectional SEM images of Sn-58Bi/ENEPIG joints with different reflow cycles: (**a**) one reflow cycle ($L_{IMC}$ 1.7 μm), (**b**) two reflow cycles ($L_{IMC}$ 2.3 μm), (**c**) three reflow cycles ($L_{IMC}$ 3.2 μm), (**d**) four reflow cycles ($L_{IMC}$ 3.8 μm) and (**e**) five reflow cycles ($L_{IMC}$ 4.3 μm).

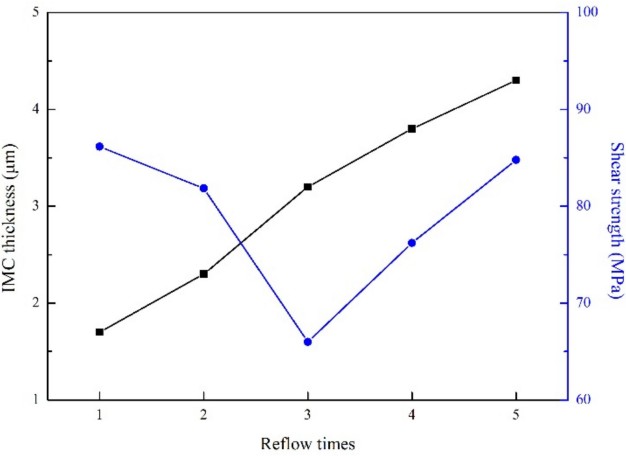

**Figure 9.** IMC thickness and shear strength of Sn-58Bi/ENEPIG joints as a function of reflow cycles.

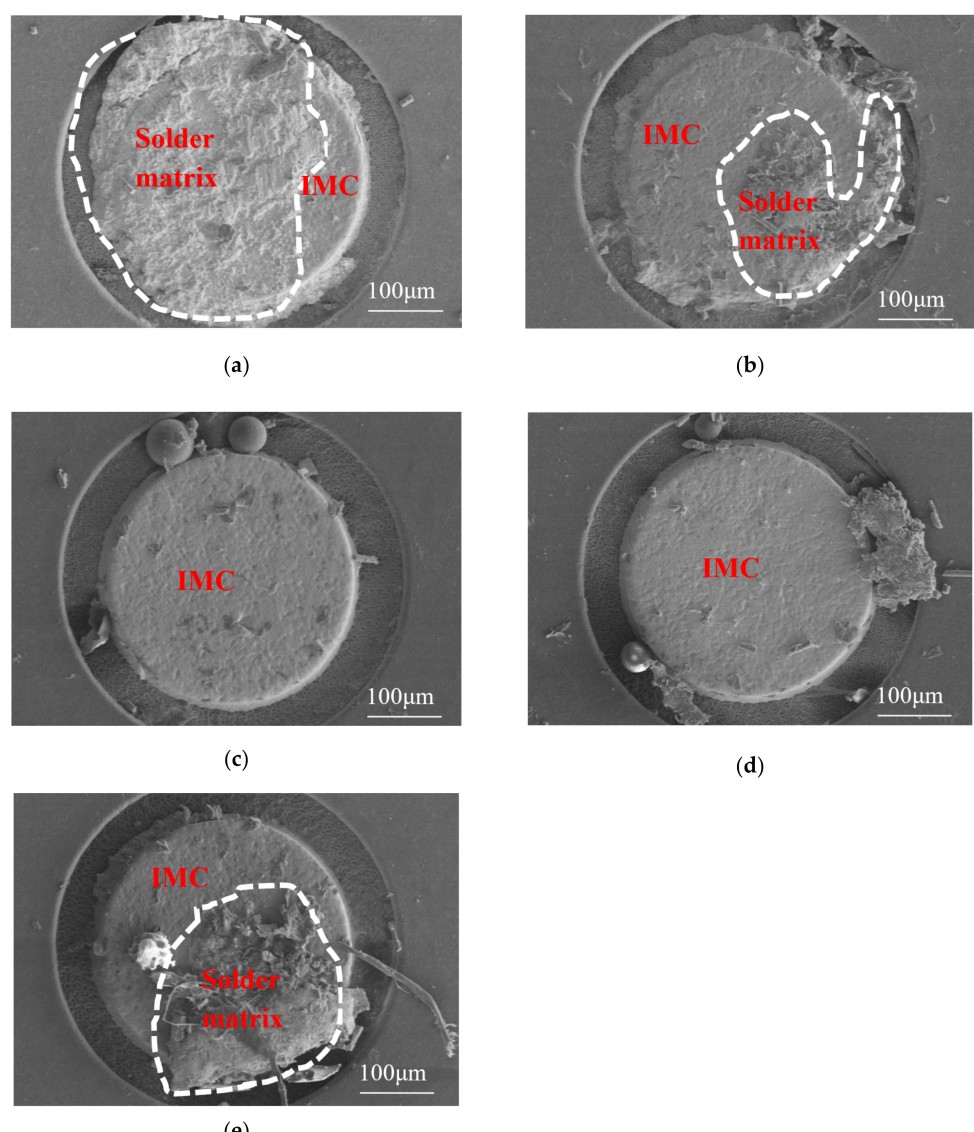

**Figure 10.** Fracture surfaces of Sn-58Bi/ENEPIG joints with different reflow cycles: (**a**) one reflow cycle, (**b**) two reflow cycles, (**c**) three reflow cycles, (**d**) four reflow cycles and (**e**) five reflow cycles.

### 3.3. Interfacial Microstructure and Mechanical Properties for Different Aging Times

Figure 11 show the surface morphology and cross-sectional SEM micrographs of the Sn-58Bi/ENEPIG joints aged at −40 °C for 100 h, 200 h, 300 h and 500 h, respectively. Figure 12 shows the relationship between IMC thickness and shear strength of Sn-58Bi/ENEPIG joints with aging time ranging from 100 h to 500 h. The thickness of the IMC layer formed at the interface aged at −40 °C, ranging from 100 to 500 h, was 1.8, 2.0, 2.2 and 2.3 μm, respectively. When the aging time was less than 300 h, the high driving force of IMC growth was provided by high temperature and a large amount of Pd, Au and Sn atoms. The thickness increased from 1.8 μm to 2.2 μm with a higher growth rate of $2 \times 10^{-3}$ μm/h. When the aging time was more than 300 h, the atom diffusion became slower because of the reduction of Pd, Au and Sn atoms. Under the same aging time, the driving force of IMC growth was lower because of the change in atomic diffusion rate. The thickness increased from 2.2 μm to 2.3 μm, with a growth rate of $0.5 \times 10^{-3}$ μm/h. In addition, there was no obvious increase in the Bi rich phase or β-Sn phase. Figure 13 shows the fracture surfaces of Sn-58Bi/ENEPIG joints. The ball shear strength of solder joints aged at −40 °C, ranging from 100 h to 500 h, was 83.76, 81.85, 75.72 and 73.49 MPa, respectively. It was found from

Figure 12 that the ball shear strength decreased by about 12.3% with the increase of aging time, which means the extension of aging time has little effect on the strength change of Sn-58Bi/ENEPIG joints and the joints show good mechanical properties at −40 °C. It can be seen from the fracture morphology that the fracture surface consisted of the solder matrix aged for 100 h, and when the aging time increased to 200 h, the fracture surface consisted of the combination of the solder matrix and the IMC layer. Therefore, when the aging time was 100 h, the fracture mode was ductile fracture. When the aging time increased to more than 200 h, the fracture mode was ductile and brittle mixed fracture. As observed from the SEM image of fracture with magnification of 5000 times, the fracture surface of Sn-58Bi/ENEPIG joints was blocked by the Bi-rich phase and β-Sn phase. When the joints were aged for 200 h, the structure of some Bi-rich phases and β-Sn phases was triangular. The fracture surface of Sn-58Bi/ENEPIG joints aged at 300 h and 500 h contained slender dendritic, large flake structures, as well as IMC particles, and the large flake structures coarsen when the aging time increased to 500 h. According to the fracture morphology, it can be concluded that the decrease in ball shear strength was mainly caused by the transition of the fracture location from the solder matrix to the IMC layer.

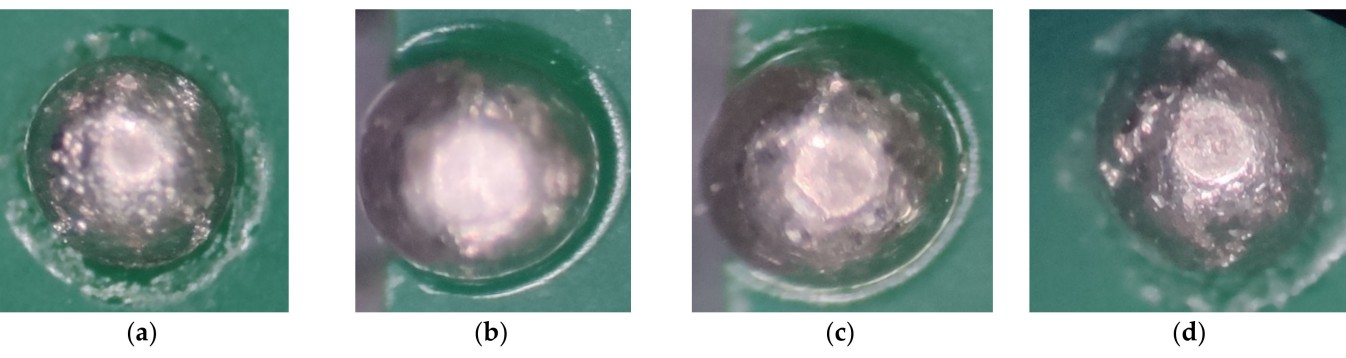

(**a**)  (**b**)  (**c**)  (**d**)

**Figure 11.** Surface morphology of solder joints with different aging times: (**a**) 100 h, (**b**) 200 h, (**c**) 300 h and (**d**) 500 h.

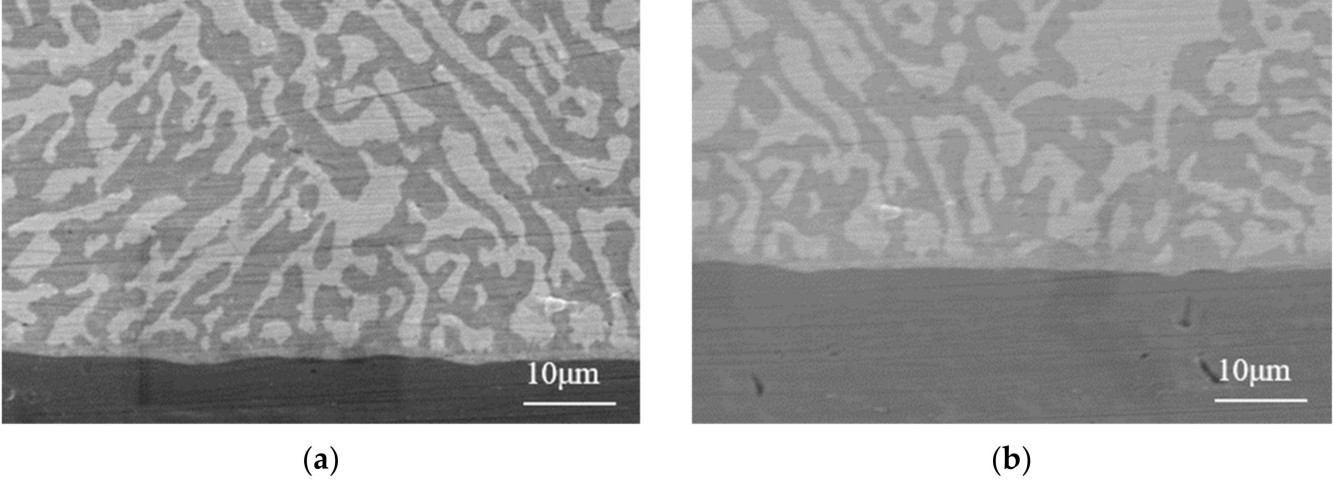

(**a**)  (**b**)

**Figure 11.** *Cont.*

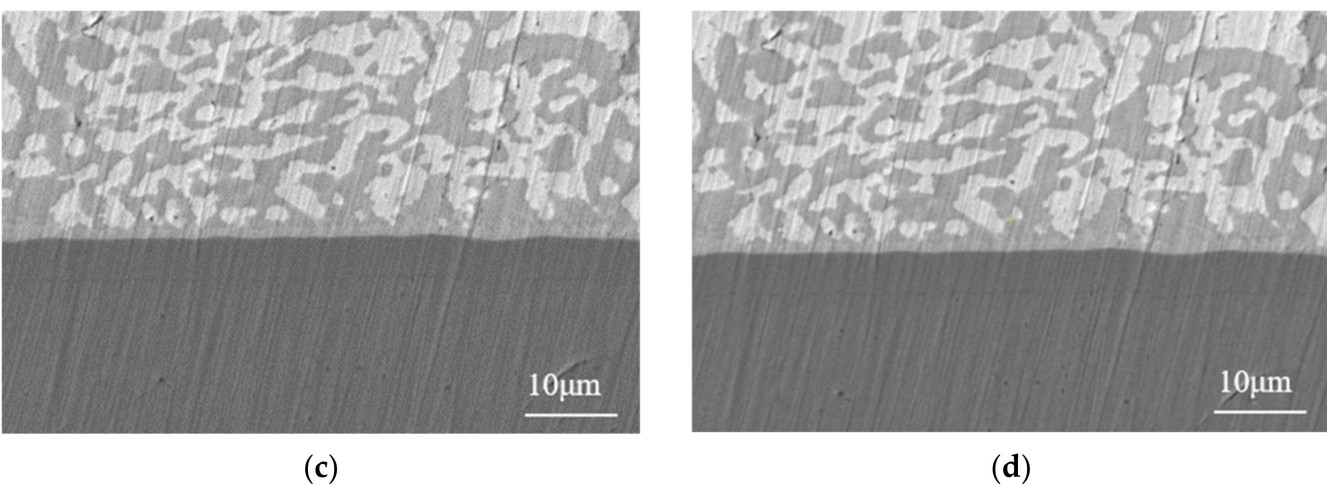

(**c**)          (**d**)

**Figure 11.** Cross-sectional SEM micrographs of solder joints after aging at −40 °C: (**a**) 100 h ($L_{IMC}$ 1.8 μm), (**b**) 200 h ($L_{IMC}$ 2.0 μm), (**c**) 300 h ($L_{IMC}$ 2.2 μm) and (**d**) 500 h ($L_{IMC}$ 2.3 μm).

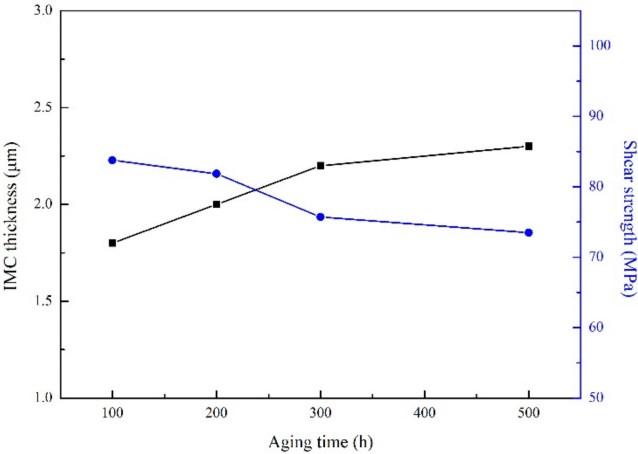

**Figure 12.** IMC thickness and shear strength of Sn-58Bi/ENEPIG joints with aging time ranging from 100 h to 500 h.

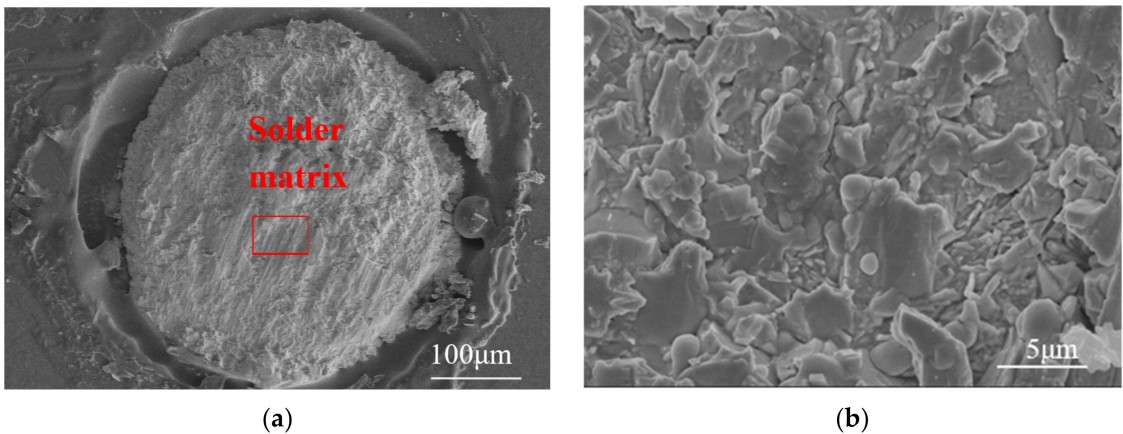

(**a**)          (**b**)

**Figure 13.** *Cont.*

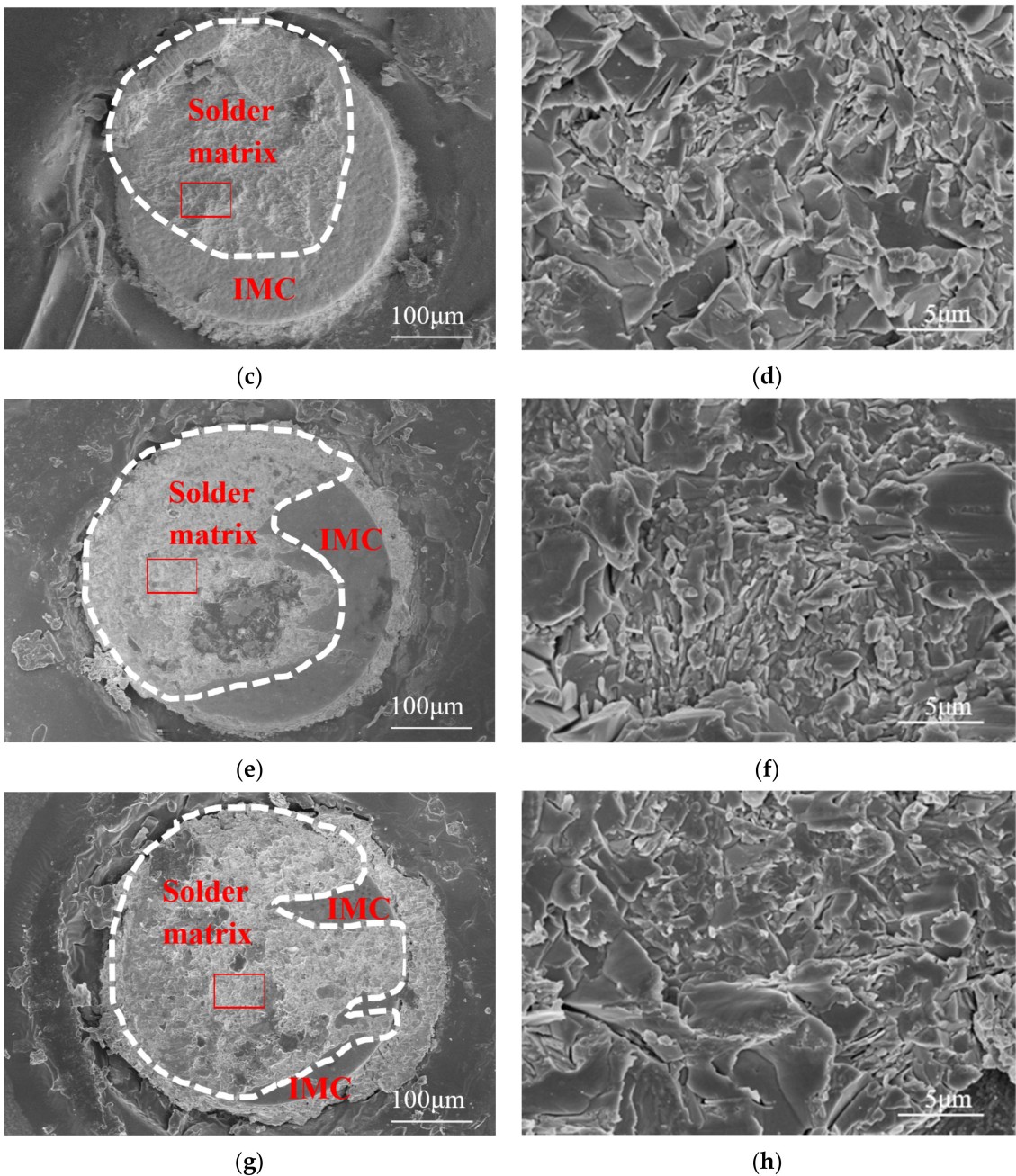

**Figure 13.** Fracture surfaces of Sn-58Bi/ENEPIG joints with various aging times: (**a,b**) 100 h, (**c,d**) 200 h, (**e,f**) 300 h and (**g,h**) 500 h. (**b**) Enlarged view in the frame of (**a**); (**d**) Enlarged view in the frame of (**c**); (**f**) Enlarged view in the frame of (**e**); (**h**) Enlarged view in the frame of (**g**).

## 4. Conclusions

In this work, the interfacial microstructure and mechanical reliability of the Sn-58Bi/ENEPIG solder joints were systematically studied. The Sn-58Bi solder ball contained an eutectic lamellar microstructure of a β-Sn solid solution and Bi-rich phase. Pd–Au–Sn IMC was formed at the Sn-58Bi/ENEPIG interface. Pd and Au elements in ENEPIG finish with Sn element in Sn-58Bi solder were formed at the Sn-58Bi/ENEPIG interface, and most of the Pd element was dissolved into the molten solder to react at the interface. The remaining thin Pd layer prevented the diffusion of the Ni layer. With the increase in reflow cycles, the surface of the solder joints was wrinkled and blackened, and the oxidation phenomenon was more serious. In addition, the IMC layer grew gradually, and by fitting the data, the relationship between reflow cycles (X) and thickness (Y, μm) of

the IMC layer was established as: Y = 0.67X + 1.05 IMC. In the ball shear test, the shear strength first decreased and then increased. After three reflow cycles, the shear strength of solder joints was the lowest. The IMC layer increased to a certain extent, and the ball shear strength slightly decreased after aging at −40 °C for up to 500 h because the low temperature limited the growth of IMC. According to the fracture morphology, the fracture mode transferred from the ductile fracture to a mixture of ductile and brittle fracture when the reflow exceeded three cycles and the aging time exceeded 200 h, owing to the fact that the fracture location transferred from the solder matrix to the IMC interface. This work provides a vital understanding of the interfacial microstructure and mechanical reliability of the Sn-58Bi/ENEPIG solder joints at low temperatures.

**Author Contributions:** Conceptualization, C.C. and C.W.; methodology, C.C., H.S. and H.X.; software, H.X. and D.N.; validation, C.C. and Q.G.; formal analysis, C.C., H.Y. and X.G.; investigation, C.C., H.S. and Q.G.; resources, C.C., H.Y. and D.N.; data curation, C.C. and C.W.; writing—original draft preparation, C.C., C.W., H.S. and H.X.; writing—review and editing, C.C., C.W., H.S., H.X. and K.B.; visualization, C.C., D.N. and Q.G.; supervision, C.C., H.Y. and Q.G.; project administration, C.C. and Q.G. All authors have read and agreed to the published version of the manuscript.

**Funding:** This research received no external funding.

**Institutional Review Board Statement:** Not applicable.

**Informed Consent Statement:** Not applicable.

**Data Availability Statement:** Not applicable.

**Conflicts of Interest:** The authors declare no conflict of interest.

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
