# Peer review of "Interfacial Microstructure and Mechanical Reliability of Sn-58Bi/ENEPIG Solder Joints"

_processes, doi:10.3390/pr10020295_

Round 1
Reviewer 1 Report
I liked the work - a good literary review, a sufficient list of literary sources, 14 literary sources over the past 5 years, a little self-citation. The technique is well laid out. A large number of research results are presented in the form of photographs and drawings. Borrowing is about 16%, but generally accepted terms and expressions, as well as literature, got into borrowing. The article will be of interest to scientists, since although the Sn37Pb eutectic solder is still an important material, due to the ingress of lead vapors into the human body and the environment, the use of lead is dangerous for people and production. The developed new eutectic solder is considered to be a quality lead-free solder, which is confirmed by the conducted research.
Author Response
Dear Editor and Review,
Thank you very much for giving us an opportunity to revise our manuscript entitled: “Interfacial microstructure and mechanical reliability of Sn-58Bi/ENEPIG solder joints” (Manuscript ID: processes-1439983). We have considered the comments carefully and made the correction accordingly. To facilitate the peer-review process, a point-by-point response to each comment is provided. Appreciate if the revised manuscript can be accepted for publication in this journal.
Yours sincerely,
Cheng CHEN
China State Shipbuilding Corporation, Limited (CSSC) 723rd Research Institute
Email: chencheng5670@163.com

Reviewer 2 Report
Dear Authors,
The paper presents the results of experiments related to the properties of solder joints influenced by the interface microstructure, formed at different reflow process profile. The problem seems to be important to design or control of quality of the solder joints.
Nevertheless, the methodology of microstructural studies as well as the correctness and repeatability of the results can be considered questionable
1/ Phase composition of the IMCs layer should be clearly identified, e.g. by means of EBSD analysis, and not only on the basis of one EDS microanalysis result in the multi-phase interface area (results of point EDS microanalysis should be presented with accuracy to 0.1%).
2/ Interface layer morphology and its thickness are not revealed on the SEM micrographs (Figs. 8, 13),
3/ Quality of the analyzed surface seems inadequate for the quantitative EDS microanalysis and the measurement of the thickness of the IMCs layer
3/ According to the data contained in the text, the thickness of the interfacial layer was measured with an accuracy of 1 nm, which is close to the limit of the imaging resolution obtained in SEM, therefore, for such values to be considered reliable, it would be necessary to use a very high microscopic magnification, perform a large number of measurements, estimate the results dispersion and the confidence interval.
Otherwise, the presented empirical relationships between the layer thickness and the number of reflow cycles cannot be considered significant. A similar remark applies to the effect of layer thickness on the ball shear strength
4/ Fracture surface topography revealed in the SEM micrographs (Figs. 11 and 16 ) is not sufficiently detailed to support considerations and conclusions included in text. The visible topography of the fracture of the entire details is heterogeneous. Thus, the enlarged areas (Fig. 16 b, d, f, h) not marked in the photos of the entire fracture (Fig. 16 a, c, e, g) cannot be correctly compared.
Author Response

(The authors gave the same response as above.)

Reviewer 3 Report
Concerning Fig1 - If the enlargement at the bottom of the image should correspond to the schematics at the top, there should be added an Sn42Bi58 layer as it is included in the dotted rectangle.
Therefore, the image should be corrected, either the bottom part or the chosen area (dotted rectangle) in the top part.
Table 1 - Perhaps bottom or lowere furnace would be a better choice to "under furnace"?
Fig.5 - Are the maps in presented in counts or wt%? at%? It should be stated in the images or in the description underneath. There is nothing visible in Au map. Is it possible to work with contrast/brightness for Bi and Pd distribution maps?
Fig.6 - What about Cu?
This is not measurement of IMCs as the marked region also includes substrate (Fig.3), this is rather EDS measurement for interface region that includes IMC layers
In "Author Contributions" there is only initial "H." for "writing—review and editing, C.C., C.W., H.," - is this correct?
Author Response

(The authors gave the same response as above.)

Reviewer 4 Report
This submission reports the microstructure and mechanical properties of Sn-58Bi/ENEPIG solder joints under various reflow and ageing conditions. Overall, it is more like an experimental report lack of specific scientific insights and discussions.
Introduction:
The authors listed previous studies on the behavior of IMC chemical reaction with ENIG and ENEPIG. However, the research gap in the field or the research question of this specific study is missing.
In addition, the background of ageing at -40 °C is required. Why -40 °C? Why hundreds of hours’ ageing?
Results and discussion:
The EDS results in Fig. 5 cannot be used to identify phases of (Pd, Au)Sn4 and Ni3Sn4. XRD result is required for correct phase identification.
The relationship between reflow cycles and IMC thickness was obtained in Fig. 9. What are the insights of this relationship in IMC nucleation and growth? https://doi.org/10.1016/j.actamat.2021.116894; https://doi.org/10.1016/j.actamat.2010.05.028; and https://www.sciencedirect.com/science/article/pii/S1359645411007683
discussed the nucleation and growth of IMC in details. More discussion is needed by comparing the relationship with literature.
What about the relationship between IMC thickness and ageing time in Fig. 14? Do they follow the same relationship as Fig. 9 and why?
The shear strengths of the joints were tested. How were the performance comparing with literature? Are they enough for real application? And what is the suggestions out of the current performances?
Figures
The current figures can be better organised. For example, Fig. 9 and Fig. 10 can be combined into one while Fig. 14 and Fig. 15 can be combined.
The scale bars in Fig. 8 and Fig. 13 a and b are missing.
Author Response

(The authors gave the same response as above.)

Round 2
Reviewer 2 Report
Dear Authors,
I have read the revised manuscript, with made modifications .
However, a few comments that are most important from the point of view of the repeatability of the results obtained in the experiment and their scientific significance remained unaccounted for.
Especially:
- You still did not present the results of the identification of the phase composition of the intermediate layer, basing this identification only on the results of the EDS microanalysis.
- The morphological characteristics of the interfacial layer have not been defined, so it cannot be clearly identified in the microstructure image. Thus its thickness cannot be measured.
Author Response

(The authors gave the same response as above.)

Reviewer 4 Report
1. "These modules may undergo multiple reflow cycles during rework or multi-gradient assembly and will be in service at -40°C for extended periods of time." References are needed here. And what does "extended periods of time" mean? Please be specific.
2. How does the chemical composition measured by EDS "1.82 at. % Ni, 14.78 at. % Pd, 74.36 at. % Sn, and 9.04 at. % Au" support the phases of (Pd, Au)Sn4. Do the math properly please, if XRD is not possible.
3. So "What are the insights of this relationship in IMC nucleation and growth in Fig. 9?"
4. What is the relationship between IMC thickness and ageing time? I would say it is more important as it shows what happens in the real applications.
5. "The shear strength of the solder joints in this manuscript is slightly higher than that reported in reference 25...." the related discussion and insights needs to be articulated specifically in the manuscript so the readers will get more information.
Author Response

(The authors gave the same response as above.)

Round 3
Reviewer 4 Report
Accept